# Rigid Protection System of Infrastructures against Forest Fires

**Gilberto Vaz** [1,2,*], **Jorge Raposo** [1,2] , **Luís Reis** [2] , **Pedro Monteiro** [2] and **Domingos Viegas** [2]

1    Coimbra Polytechnic—ISEC, Rua Pedro Nunes, 3030-199 Coimbra, Portugal
2    Department of Mechanical Engineering, University of Coimbra, ADAI, Rua Luís Reis Santos, Pólo II, 3030-788 Coimbra, Portugal
*    Correspondence: gcvaz@isec.pt

**Abstract:** The destruction caused by forest fires generates social impacts, environmental impacts, and extremely important economic impacts caused by the destruction of a wide range of infrastructures and essential goods. Therefore, as it is impossible to remove all the infrastructures from the forest and wildland–urban interface, the design of protection systems is essential. The main objective of this work is the development of a low-cost protection system, with rigid panels, requiring a simple installation, in order to protect outdoor infrastructures such as telecommunications stations, shelters, roadside enclosures, power cabinets, and other structures. A study was carried out on panels that could be used for protection in order to determine whether the protective material would be more appropriate. Taking into account the fire resistance behavior, thermal and structural properties and cost, the panels selected were the magnesium oxide fiberglass reinforced. The protection was constructed, installed on a telecommunication cabinet, and experimentally laboratory tested in a wind combustion tunnel. To collect the data InfraRed and video cameras, heat flux sensors, and thermocouples were used to determine the fire propagation, heat flux, and temperatures, respectively. The experimental data show that the low-cost protection is effective for protecting telecommunication cabinets and similar infrastructures against forest fires.

**Keywords:** forest fires; fire protection; telecommunication stations; rigid panels; wildland–urban interface



## 1. Introduction

Forest fires arise naturally from sources such as the spontaneous combustion of dry matter, under high temperatures or, most commonly, lightning strikes. Wildfire ignitions frequently arise as accidental consequences of human practices, although many have been proven to be criminal or negligent [1].

According to [2], with the current climate change, an increase in the area burned by forest fires has been observed and is expected to continue. This is obviously worrying, but there is still no proven relationship between the burned area and the number of victims caused by a certain fire. The authors of [3] introduced a case study on the climate resilience of interconnected critical infrastructures considering forest fires, which was performed in Southern France, one of the most touristic European regions. It is also one of the regions with the highest forest fire risk, which is projected to be amplified under future climate conditions. Future extreme forest fires are anticipated to impact the interconnections of electricity and transportation networks, which could further cascade to communities throughout Southern France.

Forest fires are one of the main disasters that devastate many countries every year. Fire is considered as an environmental factor acting in the Mediterranean climate, having played an obvious evolutionary role in the structure and function of Mediterranean climate ecosystems. In the aftermath of a wildfire, accelerated erosion occurs [4,5], threatening the natural regeneration process and biodiversity and biotic natural capital recovery [6,7]. Climate change and continued development on fire-prone landscapes will increase the

impact of wildfires, such as high suppression costs and loss of lives and properties [8]. The climate change effects on wildfire frequency and the devastating effects of future climate conditions with a prolonged dry and warm summer period favor the ignition and spread of wildfires. However, even in highly fire-prone ecosystems, loss of biodiversity, ecosystem function, or services after wildfire events occurring with an unnaturally high frequency, magnitude of extent, or intensity can result in land degradation or even complete transformation of the ecosystem. Besides their impact on the carbon cycle, such events, usually called megafires because of their size, reduce the amount of living biomass; affect species composition, water, and nutrient cycles; increase flood risk and soil erosion; and threaten local livelihoods through the burning of agricultural land and homes. In addition, these fires have devastating impacts on local wildlife, as animals are unable to escape from the fires or become threatened through the loss of their habitat, food, and shelter.

Not only climatic conditions, but also human activities, influence fire regimes through their effects on ignition sources and fuel characteristics in many parts of the world. The tendency of urbanization nearby or within forest ecosystems is a worldwide phenomenon that increases every year in Europe [9], but also in the United States [10], Canada [11,12], and in Chile [13] and Argentina in South America [14].

These areas are increasing what is known as the wildland–urban interface (WUI). WUI is the area where human-built structures and infrastructure abut or mix with naturally occurring vegetation types. Wildfires are of particular concern in WUI because these areas comprise extensive flammable vegetation, numerous structures, and ample ignition sources [15]. Fires at WUI are becoming increasingly hazardous for life safety and for property protection [16]. The combination of the aforementioned conditions converts wildfires to megafires. A megafire is an extraordinary fire that devastates a large area. They are notable for their physical characteristics, including their intensity, size, duration, and uncontrollable dimension, as well as their social characteristics, including suppression cost and damage to infrastructures, goods, and fatalities [17,18].

Fire behavior has two different forms of classification: normal fire behavior and extreme fire behavior [19].

According to [20], extreme fire behavior can be referred to as a very large fire or a fire that extends over a large area for a long period of time, a fire whose propagation speed or energy release rate is very high, and a fire whose rapid change in behavior encompasses a certain degree of uncertainty regarding forecasting and the risk it actually represents.

The destruction caused by these phenomena generates impacts at several levels, whether social by putting the lives of the population at risk, whether economic through the destruction of a wide range of infrastructures and essential goods, or the environment. Portugal and Greece suffered fires in 2017 and 2018, respectively, which caused more than 200 fatalities mostly among common citizens [21–27]. These fires are becoming more frequent and larger because of climate change, sometimes affecting areas that had not burned before [22,23]. Wildfires are one of the most devastating environmental hazards in Portugal, causing severe socio-economic and environmental consequences, as well several fatalities [24–30].

The forest fires occurring in Portugal, and especially those in 2017, have highlighted the importance of infrastructure protection. In 2017, until October, 356 large forest fires (burned area larger than 1000 ha) were recorded, with an estimated burned area of 520,515 hectares [29]. It was also in this year that the Pedrógão Grande fire occurred, and it is known as one of the worst fires in Portugal and Europe. This terrible catastrophe took the lives of dozens of people, injured hundreds, and caused a wave of destruction. During this fire, many infrastructures were destroyed, including telecommunications stations, poles, copper, and fiber optic cables, leading to numerous communication problems. As indicated by [30], "The failure of the communications system may have contributed to the lack of coordination of combat and rescue services, to the difficulty of people asking for help and to the worsening of the consequences of the fire". In the context of the high risk of forest fires in Portugal, this work emerged with the objective to develop a protection system applicable to infrastructures that are inserted in forest areas and wildland–urban interfaces, which can be affected by forest fires. Several accidents

caused by forest fires have been recorded in Portugal, the USA, Spain, Greece, and others, related to LPG vessels of different sizes [30–34] in WUI. These cases were closely related to wildfires in WUI and showed that LPG reservoirs can become unsafe when a WUI fire occurs. Wildland–urban interfaces are at high risk of wildfires. Defensible spaces and home ignition zones are the two main aspects that protect the lives and livelihoods of WUI [35]. Different part of the world have different rules and regulations for WUI land management.

The protection system must withstand the extremely high heat fluxes generated by forest fire fronts [36]. According to [29], the materials used for fire protection can be classified into three major groups, namely: rigid and semi-rigid materials, intumescent paints, and sprayed materials. The authors of [37] showed that the use of protective blankets made up of multiple layers of fibers allowed for the protection of infrastructures against fire.

Thin fiberglass blankets with an aluminum coating are often used for fire protection due to their capacity to withstand extreme incident radiant and convective heat fluxes [38]. Recently, researchers applied these materials for the protection of infrastructures against wildfires [39]. However, the application of this type of protection only becomes viable when the exposure time is relatively short, as for longer exposures, deterioration of the protection becomes evident.

The work developed emerges as a continuation of previous studies [40] in which a fiberglass blanket coated with aluminum was used in order to protect a telecommunications cabinet. This closed protection, in addition to being effective at protecting against fire, also showed that the temperature inside the cabinet increased, even when there was no exposure to fire. In addition, this type of protection is not suitable for lengthy outdoor exposure, as it deteriorates easily, namely by the wind and solar radiation. Thus, the author left open the possibility of using another type of protection. As forest fires are a transient phenomenon, thermal inertia of protection is important in order to reduce the temperature increase of the protection and air inside the cabinets or enclosures. So, in this work, rigid panels will be used and applied to cabinet stations to test fire resistance. The novelty of the protection is shown by its effectiveness for the protection of telecommunication cabinets and other similar infrastructures against forest fires, even in wind driven fires. The protection avoided the high temperatures in the cabinet. Without protection, a very expensive and critical system can be easily damaged by the fire front of a wildfire. Another great advantage of this protection is its low cost of material and the reduced labor for installation, allowing for daily work on the cabinet if necessary.

## 2. Materials and Methods

### 2.1. Protective Device

The first step of the work was the study and selection of the most appropriate material panel for protection. The panels to be used in the protection must be impact resistant, fire-resistant, resistant to climatic adversities, and must have low thermal conductivity and high thermal inertia to withstand outdoor ambient conditions and forest fire transient effects. Several panels were identified and compared, as shown in Table 1. The selected panel was the MAGOOX Board panel with 9 mm thickness [41], as the boards were produced with thicknesses of 4 to 18 mm, with the thickness of 9 mm being an average value that will not turn the protection into a heavy structure. This material meets all the requirements for the protection material and it is non-toxic, has a low environmental impact, and is available from construction material suppliers at a low price. The current approximate European price of the selected panel was 12 €·m$^{-2}$. This panel was composed of fiberglass reinforced magnesium oxide and has a fire resistance of 60 to 90 min. Generally, forest fire fronts have a residence time of no more than 15 to 20 min, and they often reach very high propagation speeds [42].

**Table 1.** Commercial panels identified for the study.

| Commercial Name | DuraSteel | Promatec-XW | WeatherKem | SpeedPanel | MAGOOX Board | TriplacM |
|---|---|---|---|---|---|---|
| Composition | 2 perforated steel plates and fiber reinforced cement core | Gypsum board with mat reinforcement | Mixture of cement, cellulose fibers, and silicon-based binders | Cement core and galvanized steel cladding | Magnesium oxide reinforced with fiberglass mesh | Magnesium, fiber reinforced, and other refractory materials |
| Thickness [mm] | 9.5 | 15 | 6–18 | 51; 64; 78; | 4–18 | 24–30 |
| Fire resistance [min] | 240 | ≤60 | are completely non-combustible | Varies with thickness (60; 90; 120) | 60–90 | 180–240 |
| Resistance to climatic adversities | Yes | Yes | Yes | Yes | Yes | Yes |
| Impact resistance | Yes | Yes | Yes | Yes | Yes | Yes |
| Thermal conductivity (20 °C approx.) [W/m.K] | 0.55 | 0.264 | 0.24 | Varies with cement densities | 0.213 | 0.29 |

The protection was constructed and installed on a telecommunication cabinet (cabinet dimensions of 1.60 m height, 0.70 m width, and 0.56 m depth), surrounding the cabinet without gaps, as shown in Figure 1. Only the upper part was left open, to allow for the convection cooling of the cabinet. The panels were installed, fixed to the cabinet with metallic screws and washers, creating a space air gap of 5 cm, allowing for normal work activity, particularly the door opening.

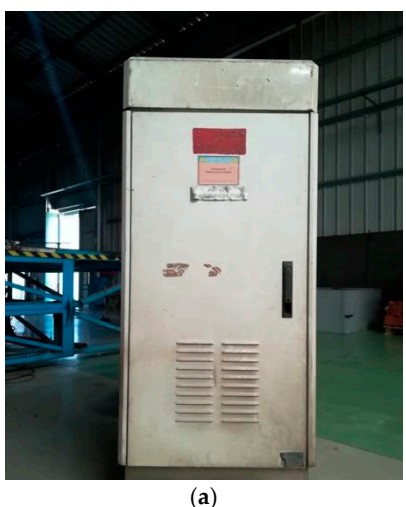
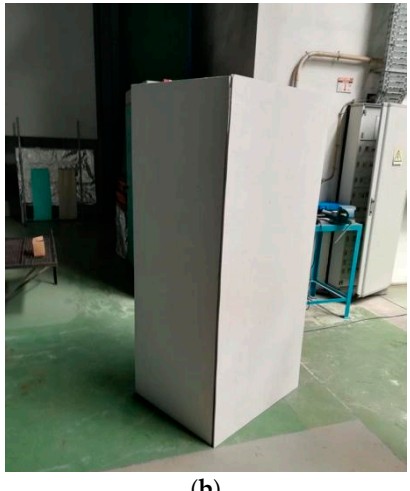

(**a**)　　　　　　　　　　　　　　　　(**b**)

**Figure 1.** (**a**) Telecommunication cabinet and the (**b**) protected telecommunication cabinet.

## 2.2. Laboratory Tests

To study the thermal behavior of the cabinet and its protection, they were subjected to a series of tests carried out at LEIF-ADAI facilities using the Combustion Tunnel 3 (CT3), which has two 35 kW fans that can generate air speeds of up to 8 m·s⁻¹. This tunnel had dimensions of 8 m length, 6 m width, and sidewalls 2 m height. The experimental apparatus

is shown in Figures 2 and 3. The fuel bed area was defined by a fixed length and width of 4 m, corresponding to 16 m$^2$, with a fuel load of 1.5 kg·m$^{-2}$, on dry basis, of typical Mediterranean shrubs [43] composed of a mixture of *Erica umbelatta*, *Erica australis*, *Ulex minor*, and *Chamaespartium tridentatum*. The moisture content of the fuel varied in the range of 12.6 to 15.1 (wet basis). The wind speed varied in the range from 0 to 3 m·s$^{-1}$. These are standard values used in fire experiments in wind tunnels, according to [43,44] The flow over the floor of the tunnel is of a boundary layer type with a reference velocity Uo, which is measured at the center of the working section floor and 0.5 m above the ground. This corresponds to a freestream at 10 m height standard wind readings of 1.5 xUo. The fire experiments were performed with the following values of Uo: 0,1, 2, and 3 m·s$^{-1}$, which corresponds to winds at 10 m height of 0, 1.5, 3.0, and 4.5 m·s$^{-1}$ (0, 5.4, and 10.8 to 16.2 km·h$^{-1}$).

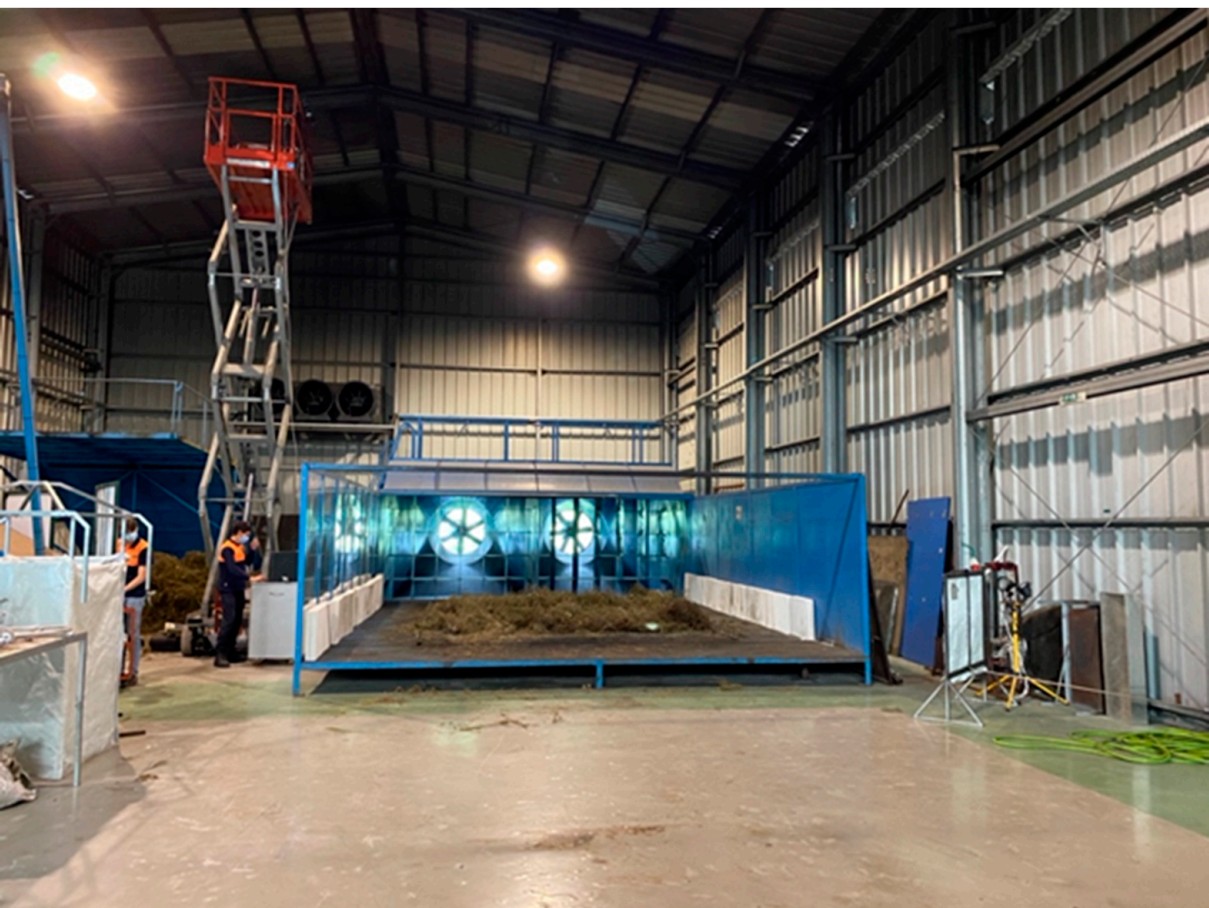

**Figure 2.** Combustion tunnel CT3.

The existence or not of protection and the existence of sidewalls (which avoid lateral air entrainment, promoting the arrival of a structured front, without an edge effect) were tested. The list of tests performed and the respective variables are presented in Table 2 and they were performed in random order. These types of tests are very time consuming and require a lot of laboratory equipment. Preliminary tests were carried out to assess the need to repeat the tests for the purpose of evaluating the effectiveness of the thermal protection. As the recordings of the temperatures and heat fluxes were consistent, this showed a stable evolution without significant random oscillations. A single test was performed for each type of test.

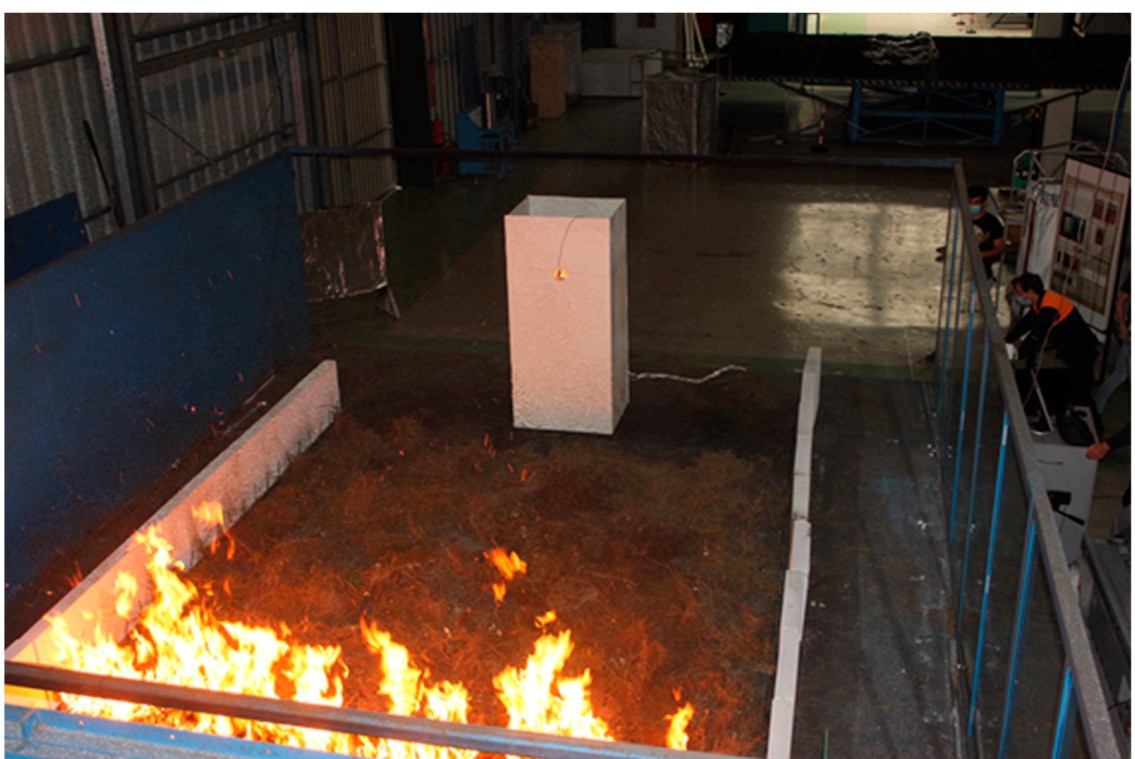

**Figure 3.** General view of the tests.

**Table 2.** Conditions of the experimental tests performed.

| Ref. | Wind Speed (U [m·s⁻¹]) | Protection | Sidewalls |
|------|------------------------|------------|-----------|
| **01** | 0 | Yes | No |
| **02** | 1 | Yes | No |
| **03** | 3 | Yes | No |
| **04** | 0 | Yes | Yes |
| **05** | 1 | Yes | Yes |
| **06** | 3 | Yes | Yes |
| **07** | 0 | No | No |
| **08** | 1 | No | No |
| **09** | 0 | No | Yes |
| **10** | 1 | No | Yes |
| **11** | 3 | No | Yes |

In the tests, an InfraRed FLIR Camera SC660 (640 × 480 image resolution; sensitivity <30 mK; −40 °C to 1500 °C standard temperature range; 2% or 2 °C accuracy; 1–8 times continuous zoom), a heat flux sensor (Hukseflux sensor IHF01), and five sheathed K-type thermocouples of inconel with 1 mm diameter were used. The referred equipment was used to determine the fire propagation, heat flux, and temperatures. Additionally, for all of the tests, two optical video cameras were used. One video camera, Sony FDR-AX53 (4 K Ultra HD (3840 × 2160) recording; Sensor type—1/2.5 type (0.28 in) Exmor R CMOS; Lens type—ZEISS Vario-Sonnar T; Optical zoom—20×), was placed on a lift platform and the other video camera, Sony HXR-NX30E (Full HD; Sensor type—1/2.88-inch type Exmor R CMOS; Lens type—Ultra wide-angle 26.0 mm Carl Zeiss Vario-Sonnar T; Zoom—10× (optical), 17× Extended Zoom), was placed at ground level. The experimental setup is shown in Figure 4.

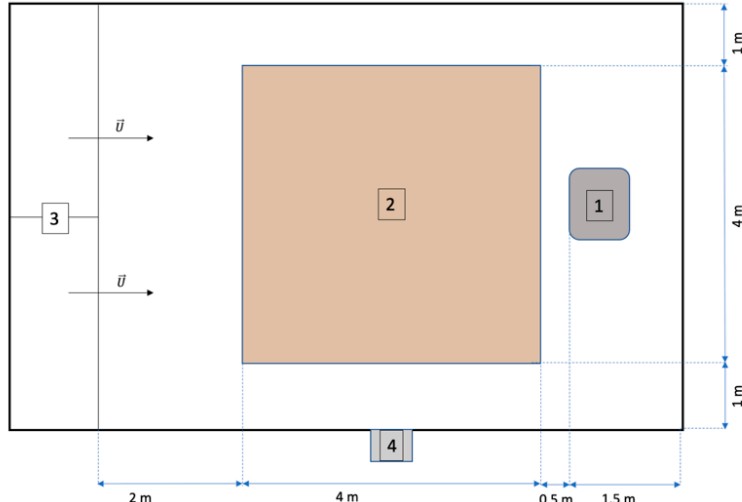

**Figure 4.** Experimental setup (1—telecommunication cabinet, 2—fuel bed, 3—wind tunnel fans, 4—infrared and video cameras).

To acquire and record data, an acquisition board NI chassis cDaq 9174, a thermocouple board 9213, a voltage board 9211, and a computer with the signalExpress program with an acquisition frequency of 1 Hz were used. The thermocouples and heat flux sensor were installed in the following positions:

Thermocouple 1($T\_1$): placed inside the cabinet (roughly in the center, about 1.30 m from the ground);

Thermocouple 2 ($T\_2$): placed on the inner front surface of the protection (about 1.30 m from the ground) and in the tests carried out without protection, which was placed on the side of the cabinet;

Thermocouple 3 ($T\_3$): placed on the outer front surface of the protection or cabinet (in absence of the protection), at about 1.30 m from the ground;

Thermocouple 4 ($T\_4$): placed behind the cabinet about 1 m away in order to be able to assess the downstream air temperature;

Thermocouple 5 ($T\_5$): placed on the sidewalls (when applicable);

Heat flux sensor: placed on the outer front surface of the protection (or of the cabinet, respectively, if testing with or without protection), approximately 1.30 m from the ground. The flux and temperature values at the various points of the shield protection and cabinet were taken directly from the thermocouple data and were converted into Microsoft Excel®.

The heat flux data were obtained making a correction to the value read by the heat flux sensor, to take into account the temperature dependence of the sensitivity. The heat flux is given by Equation (1), referred to in the heat flux sensor user manual.

$$\Phi = U/(S \times (1 - 0.0005 \times (T - 20))) \tag{1}$$

In Equation (1), $\Phi$ [W/m$^2$] is the heat flux, U [V] is the output voltage that is read directly from the signal generated by the heat flux sensor, S [V/(W/m$^2$)] is the sensitivity of the sensor at 20 °C, and and T [°C] is the temperature read on thermocouple of the sensor. The sensitivity of the sensor is available in the sensor calibration certificate and takes a value of $9.83 \times 10^{-9}$ V/(W/m$^2$) with a calibration uncertainty of $\pm 0.98 \times 10^{-9}$ V/(W/m$^2$). As referred to in the calibration certificate, this calibration uncertainty corresponds to the expanded uncertainty with a coverage factor k = 2, and defines an interval estimated to have a level of confidence of 95%.

The apparatus expanded uncertainty of the heat flux measurements, for a confidence level of 95%, was calculated according to [45], taking into account the main sources of uncertainty, namely: the calibration uncertainty of the heat flux sensor, uncertainty due to the input noise error of NI 9211 board, and uncertainty due to the NI 9211 board sensitivity.

Uncertainty due to the systematic error of the NI 9211 board was neglected as the systematic error was adequately corrected. The uncertainty due to the accuracy of the thermocouple of the sensor was neglected as the order of magnitude was lower in comparison with the other uncertainties. Consulting the NI 9211 board specifications [46], a 1 µV input noise and 1 µV board sensitivity were considered.

The apparatus expanded the uncertainty of the temperature measurements, for a confidence level of 95%, and was calculated according to [44,45], taking into account the main sources of uncertainty, namely: uncertainty due to the thermocouple accuracy, uncertainty due to the NI 9213 board error when connected to k type thermocouples, and uncertainty due to the NI 9213 board sensitivity. Consulting the thermocouple specifications [47,48], the K type thermocouple accuracy was calculated using the function max (1.5 °C; 0.004 ∗ temp [°C]). Consulting the NI 9213 board specifications [48], the maximum error of the NI 9213 board at room temperature when connected to k type thermocouples, for our temperature range (20 °C < T < 800 °C), was 1.5 °C and the board sensitivity was 0.25 °C.

## 3. Results

Figure 5 shows the evolution of the heat fluxes for three representative cases where the wind speeds conditions were changed from 0 to 3 m·s$^{-1}$, conducting to different fire intensities and spread rates. Figure 5 also shows the apparatus expanded uncertainty intervals of the heat flux measurements. The fire intensity was perceived by the maximum heat fluxes measured, of approximately 9, 5, and 1 kW·m$^{-2}$, for tests 03, 02, and 01, respectively. As shown clearly in Figure 5, the heat fluxes' history depended strongly on the fire spread rate (shorter burning times for higher spread rates, as expected). The heat flux measured fluctuations also resulted from the typical behavior of natural fires [49].

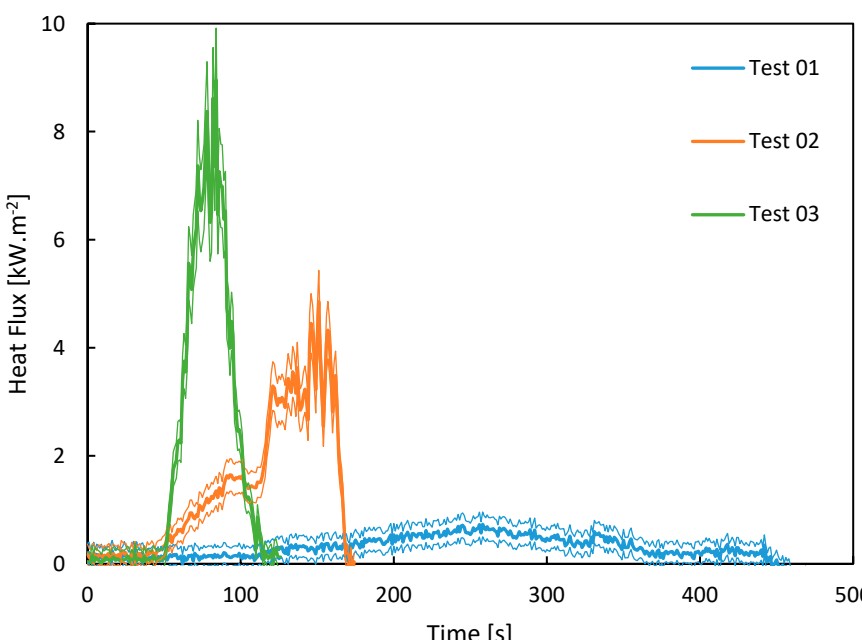

**Figure 5.** Evolution of heat fluxes for tests 01, 02, and 03, with different wind speeds of 0, 1, and 3 m·s$^{-1}$, respectively.

Figures 6 and 7 show the temperature evolution in two tests performed with the protection applied to the telecommunications cabinet and 3 m·s$^{-1}$ wind speed (highest wind tunnel speed, which is the most critical situation tested and corresponds to the typical ground level wind speeds of intense forest fires). Regarding the temperature inside the cabinet (thermocouple T_1), it was possible to verify that the maximum temperatures did not exceed 30 °C, proving that the use of the protection under study was capable of protecting the cabinet against the high temperatures and heat fluxes of a forest fire.

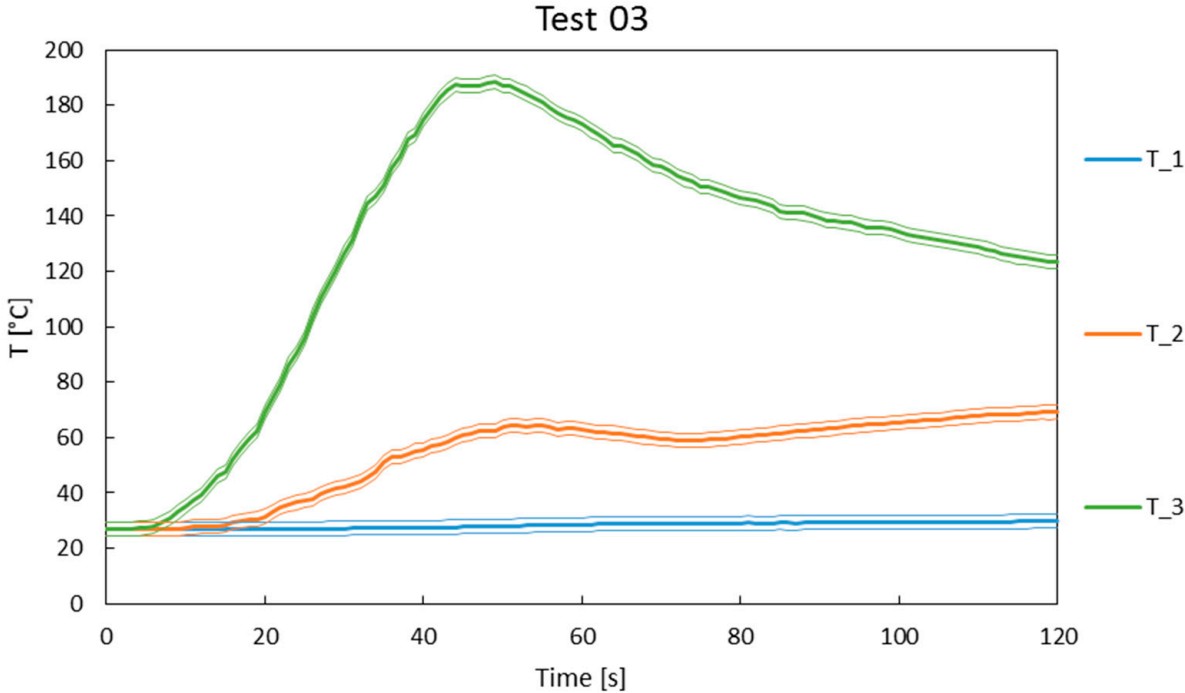

**Figure 6.** Evolution of temperatures in test 03.

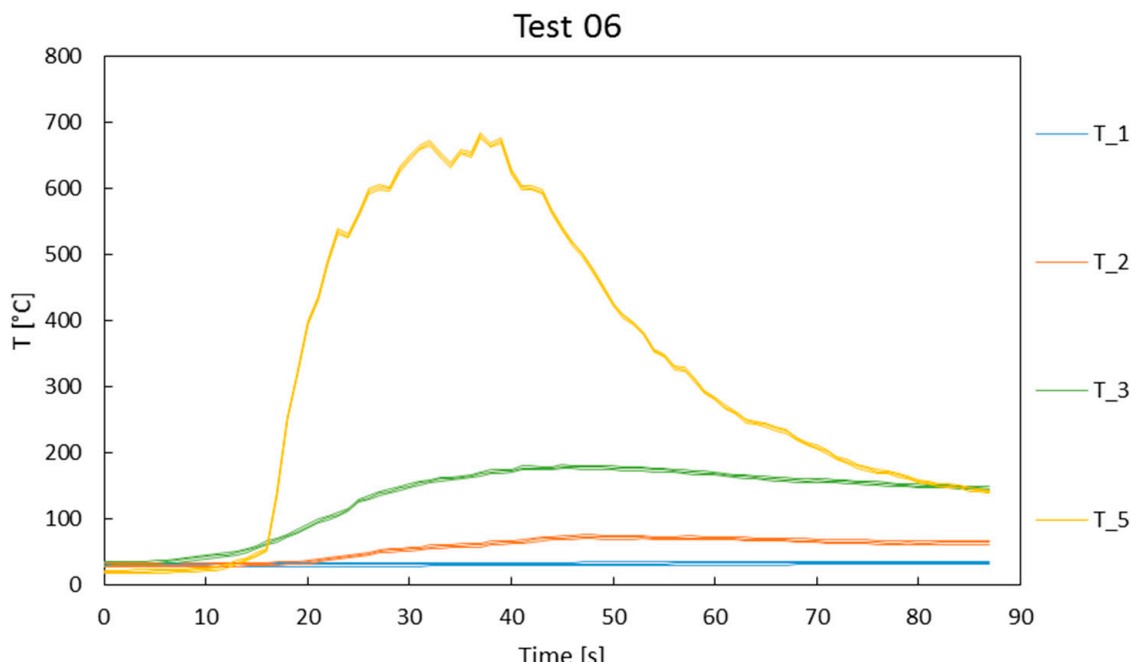

**Figure 7.** Evolution of temperatures in test 06.

Figure 8 shows the temperature evolution in the test performed without protection applied to the telecommunications cabinet and with 3 m·s$^{-1}$ wind speed. For this case, the evolution of temperatures was quite similar to that observed previously. However, there was a considerable difference with regard to the temperature registered inside the cabinet, reaching a value above 60 °C (considered critical for the normal operations of communication equipment).

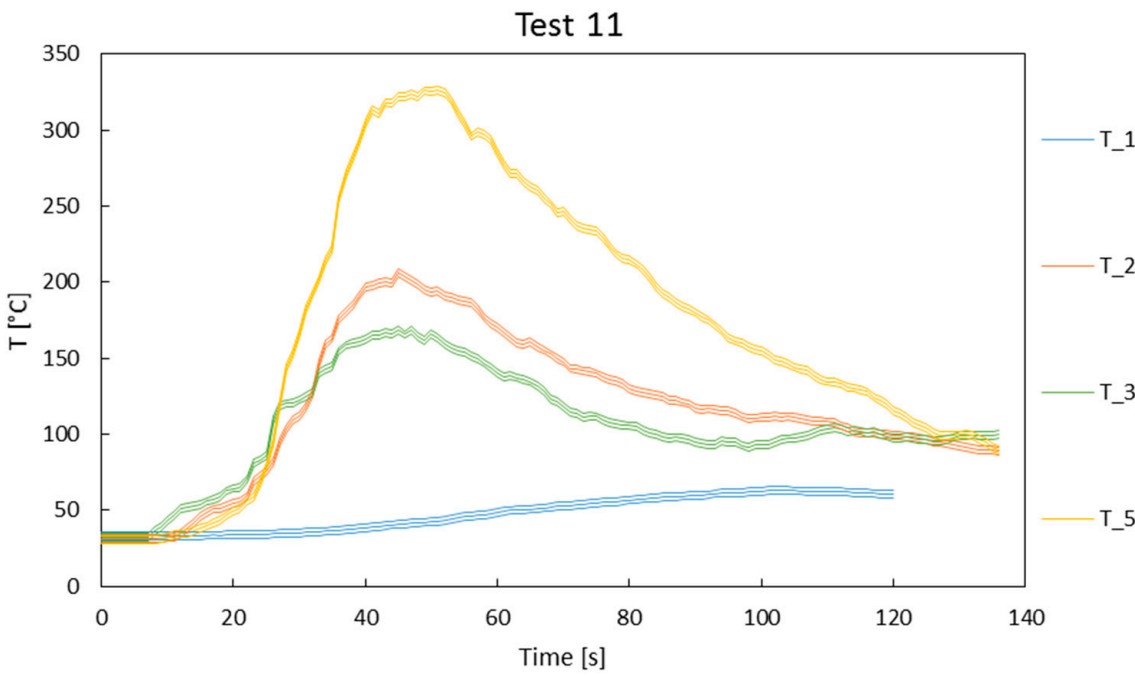

**Figure 8.** Evolution of temperatures in Test 11.

Figures 6–8 also show the apparatus expanded uncertainty intervals of the temperature measurements.

Figures 9 and 10 show the fire front shape evolution for 5 s time steps for the tests with $3 \, \text{m·s}^{-1}$ wind speed, protection installed, without sidewalls and with sidewalls (tests 03 and 06, respectively).

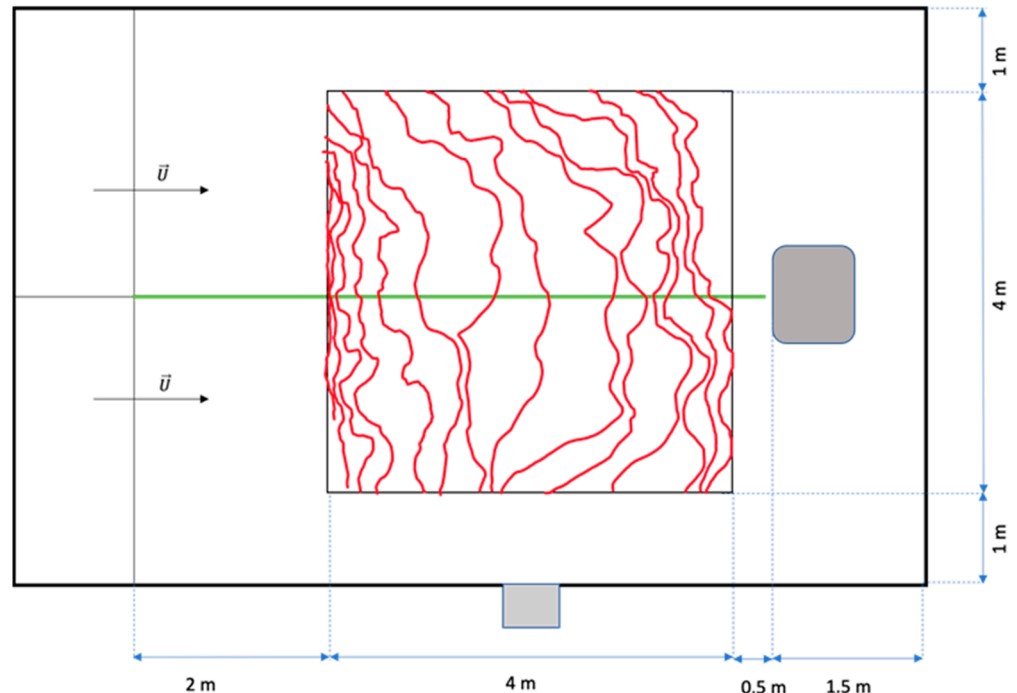

**Figure 9.** Fire front shape evolution for a 5 s time step—test 03 (without sidewalls).

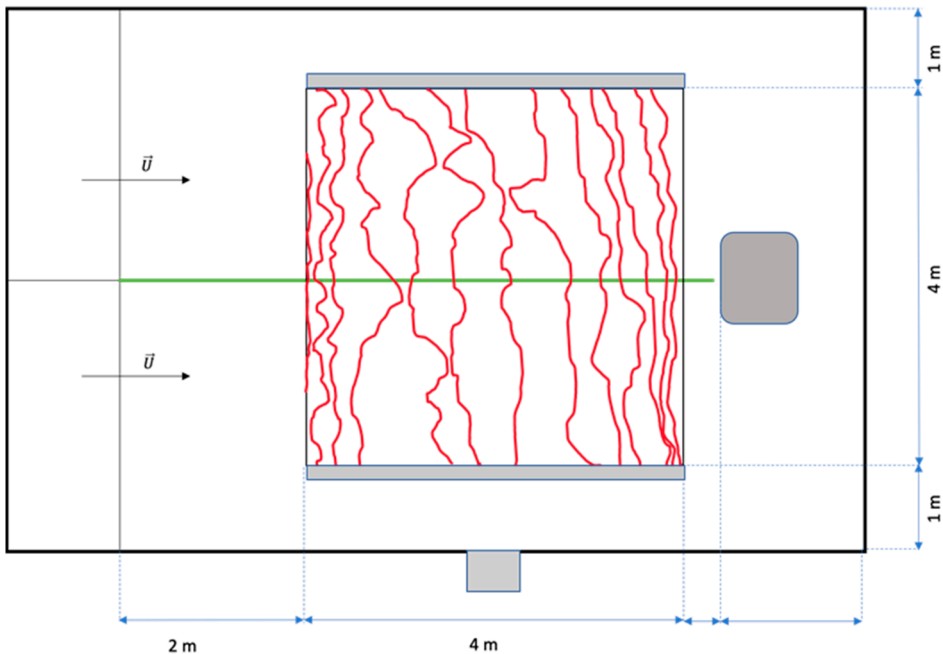

**Figure 10.** Fire front shape evolution for a 5 s time step—test 06 (with sidewalls).

To assess the protection efficiency of the system, a non-dimensional parameter called protection effectiveness (PE) was calculated. Equation (2) gives us PE as the ratio between the difference of the external surface temperature (T3) and the inner temperature of the cabinet (T1) and the maximum possible difference in relation to the ambient temperature.

$$PE = (T3 − T1)/(T3 − Tamb) \tag{2}$$

Table 3 presents a summary of the experimental results.

**Table 3.** Main results of the tests performed.

| Test Reference | Protection | Wind Speed [m·s$^{-1}$] | Side Walls | Max Dimensionless Rate of Spread | Maximum Fireline Intensity [MW·m$^{-1}$] | T3 [°C] | T1 [°C] | PE | Max Heat Flux [kW·m$^{-2}$] |
|---|---|---|---|---|---|---|---|---|---|
| 01 | Yes | 0 | No | 1.95 | 0.59 | 90.63 | 24.10 | 0.99 | 0.85 |
| 02 | Yes | 1 | No | 7.73 | 2.34 | 158.90 | 26.97 | 0.99 | 4.87 |
| 03 | Yes | 3 | No | 18.10 | 5.48 | 188.63 | 29.72 | 0.98 | 9.07 |
| 04 | Yes | 0 | Yes | 1.68 | 0.51 | 61.87 | 26.37 | 0.99 | 1.14 |
| 05 | Yes | 1 | Yes | 5.71 | 1.73 | 144.86 | 29.02 | 1.00 | 2.79 |
| 06 | Yes | 3 | Yes | 14.96 | 4.53 | 179.24 | 33.72 | 0.98 | 9.12 |
| 07 | No | 0 | No | 1.65 | 0.50 | 113.51 | 30.20 | 0.95 | 1.43 |
| 08 | No | 1 | No | 6.80 | 2.06 | 70.37 | 33.46 | 0.87 | 0.31 |
| 09 | No | 0 | Yes | 1.63 | 0.49 | 108.89 | 34.60 | 0.95 | 1.54 |
| 10 | No | 1 | Yes | 4.56 | 1.38 | 196.22 | 42.64 | 0.93 | 0.34 |
| 11 | No | 3 | Yes | 19.85 | 6.01 | 168.67 | 63.03 | 0.77 | 9.16 |

## 4. Discussion

Figures 6 and 7 show that for the higher laboratory wind speed, the heat absorbed by the protection was significant, even in these laboratory tests, as the outer front surface temperatures of the protection reached 180–190 °C (thermocouple T_3). Figure 8 shows that, without protection, the outer front surface of the cabinet also reached approx. 170 °C (thermocouple T_3). This lower temperature in comparison with the other tests with protection installed (tests 03 and 06) could be explained as a consequence of the higher thermal conductivity of the metallic cabinet material.

Figures 6 and 7 also show clearly the thermal behavior of the protection for the case of a higher laboratory wind speed. It can be observed that the inner front surface temperatures of the protection (thermocouple T_2) reached 60–70 °C. Comparing these temperatures to the outer front surface temperatures of the protection (thermocouple T_3), 180–190 °C, as referred above, the effect of the low thermal conductivity and high thermal inertia of the protection is clear, as expected.

Figure 8 shows that the heat transfer problem studied was clearly a transient phenomenon, as the temperature inside the cabinet (thermocouple T_1) increased, even when the other temperatures were already decreasing. This is why it is important to install a protection with a high thermal inertia.

Figures 9 and 10 and Table 3 show that the sidewalls are important. Comparing Figures 9 and 10, the inexistence of side walls was observed (test 03), allowing for lateral air entrainment with a significant edge effect, which generated a more curved fire front, with a higher rate of spread along the symmetry line of the fuel bed. On the contrary, when the sidewalls were installed (test 06), the lateral air entrainment was reduced, the fire front was more linear, and the rate of spread along the symmetry line of the fuel bed was slightly lower. The values for the maximum dimensionless rate of the spread are presented in Table 3 (test 03—18.1; test 06—14.96).

Table 3 shows that the protection was effective, as the maximum temperature inside the cabinet remained below 30 °C. Even under high fireline intensities (tests 03 and 06), the temperature evolution in two tests performed with the protection applied to the telecommunications cabinet and for $3 \text{ m} \cdot \text{s}^{-1}$ wind speed (highest wind tunnel speed, which is the most critical situation tested and corresponds to the typical ground level wind speeds of intense forest fires). These results are better when compared with the ones obtained in similar tests using a non-rigid protection through thin fiberglass blankets with an aluminum coating tested in the same cabinets by [44].

Table 3 also shows that the PE parameter was very high (between 0.98 and 1) for all of the experiments with the protection installed (tests 01 to 06). For the tests with no protection installed (tests 07 to 11), the calculated PE was lower, varying from 0.77 to 0.95. In both series of tests, PE decreased as the fireline intensity increased, as expected. For the particular case of test 11, with no protection installed, the inner temperature of the cabinet reached a value above 60 °C, which was considered critical for the normal operations of this type of communication equipment.

## 5. Conclusions

This work focused on the testing and implementation of a protection of a telecommunication cabinet against forest fires. Several lab experiments were made in order to assess the performance of the protection system. The data obtained in the experimental tests show that this simple, low-cost protection is effective for the protection of telecommunication cabinets and other similar infrastructures against forest fires. This protection avoided high temperatures entering the cabinet. The protection was effective, as the maximum temperature inside the cabinet remained below 30 °C, even under high fireline intensities.

The temperature obtained inside the cabinet, without protection, reached a value above 60 °C. This value is considered to be critical for the normal operations of this type of communication equipment.

This shows that without a protection system, a very expensive and critical system can be easily damaged by the fire front of a wildfire.

Another great advantage of this protection is its low cost of material and labor for installation. In this case, the protection was built and installed at a cost of approximately EUR 150, increasing the protection of an asset that costs thousands of euros.

Installing this type of protection does not compromise system operations at all. The access for maintenance or repairing is maintained, as the plates are integrated in the movement of the openings of the cabinet.

As future work, a physical and mathematical model shouls be developed to accurately reproduce the heat transfer phenomena and temperature variation in the protection and cabinet, according to the incident heat flux, that also allows for studying the effects of the material, thickness, and shape of protection.

In addition, it will be important to test this type of protection under real fires, at a field scale.

Additionally, the study of the combination of water sprinkling mechanisms that increases the humidity level of the forest fuel in the vicinity of the barrier, preventing the igniting and the fire from passing to the barrier may be addressed.

**Author Contributions:** Conceptualization, G.V. and D.V.; Data curation, P.M.; Investigation, G.V., J.R., L.R. and P.M.; Project administration, J.R., L.R. and D.V.; Writing—original draft, G.V., J.R., L.R. and D.V. All authors have read and agreed to the published version of the manuscript.

**Funding:** The work reported in this article was carried out in the scope of the FirEUrisk project—Developing a Holistic, Risk-Wise Strategy for European Wildfire Management, which received funding from the European Union's Horizon 2020 research and innovation program under the grant agreement no. 101003890, as well as the project McFire (PCIF/MPG/0108/2017), SmokeStorm (PCIF/MPG/0147/2019) and SafeFire PCIF/SSO/0163/2019, supported by the Portuguese National Science Foundation.

**Data Availability Statement:** Available on request to authors.

**Acknowledgments:** The support given by Nuno Luís for the performance of the laboratory experiments is gratefully acknowledged.

**Conflicts of Interest:** The authors declare no conflict of interest. The funders had no role in the design of the study; in the collection, analyses, or interpretation of data; in the writing of the manuscript, or in the decision to publish the results.

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
