# Peer review of "Rigid Protection System of Infrastructures against Forest Fires"

_fire, doi:10.3390/fire5050145_

Round 1

Reviewer 1 Report

Dear authors!

I have reviewed your article.

I think this article will be interesting for fire science community.

But there are some disadvantages were marked in article at present version of manuscript.

Introduction

You considered complex problem of protection infrastructures agains the forest fires. I found only five references in Introduction section. Sorry, but this is totally unacceptable for research article. You must provide background fully describing this topic within Introduction section. I work within forest fire influence on infrastructure too. I estimate suitable quantity of relevant articles for Background about 50 new references. You should consider subtopics devoted to different forest fire characteristics, hazardous materials characteristics, mathematical and experimental modelling of forest fire influence on different objects like houses, industrial facilities, infrastructure constructions. More over, you should consider regional studies on WUI fires. Please, completely rework your version of Introduction.

It is unrespectable to fire science community to consider five references in Introduction.

Materials and Methos

One remark to Figure 1.

Please clarify, how operators can serve protected cabinet in everyday work activity?

Figure 2 should be large. This is more suitable for readers.

Table 2. Please clarify, why did you choose wind speed in the range from 0 to 3 meter per second? Could you provide any supporting references or considerations?

Please, provide characteristics of cameras Sony FDR-AX53 and Sony HXR-NX30E. This is more suitable for readers.

Could you provide supporting references or consideration for your quantuity and positions of thermocouples. It seems to me you need more thermocouples to take into account spatial inhomogenity of forest fire front.

Results

I did not find confidential intervals in the figures with results. Sorry, but you must provide information about uncertainty and error analysis. You should analyze both apparatus and random uncertainty. At present time your results looks like poor soundness figures. Please, provide description of uncertainty analysis procedure in materials and methods section and confidential intervals in all Figures with results.

Article is unacceptable for publication in Fire journal at present time.

I suggest major revision. 

Author Response

Responses to Reviewer 1

Manuscript ID: 1832472

Title: Rigid protection system of infrastructures against forest fires

Authors: Gilberto Vaz * , Jorge Rafael Raposo , Luís Duarte Reis , Pedro Monteiro, Domingos Xavier Viegas

The authors would like to thank the reviewer for the constructive comments. The comments and concerns raised have been duly addressed in this revised version of the paper.

Responses to reviewer review are detailed described below. In the revised manuscript, the revised sentences are inserted with track changes.

Reviewer#1

Dear authors!I have reviewed your article.I think this article will be interesting for fire science community.But there are some disadvantages were marked in article at present version of manuscript.

  • You considered complex problem of protection infrastructures agains the forest fires. I found only five references in Introduction section. Sorry, but this is totally unacceptable for research article. You must provide background fully describing this topic within Introduction section. I work within forest fire influence on infrastructure too. I estimate suitable quantity of relevant articles for Background about 50 new references. You should consider subtopics devoted to different forest fire characteristics, hazardous materials characteristics, mathematical and experimental modelling of forest fire influence on different objects like houses, industrial facilities, infrastructure constructions. More over, you should consider regional studies on WUI fires. Please, completely rework your version of Introduction.

It is unrespectable to fire science community to consider five references in Introduction.

Answer: Thank you. We agree with your comment and we revised the text. The following text and references were added according the requirements of reviewer 1 and 2 (lines 30-107 of revised document):

Fire is considered as environmental factor acting in the Mediterranean climate, having played an obvious evolutionary role in the structure and function of Mediterranean climate ecosystems. In the aftermaths of the wildfire accelerated erosion occurs (Stefanidis et al. 2022a, Efthimiou et al. 2018) threaten the natural regeneration process and biodiversity and biotic natural capital recovery (Stefanidis et al. 2022b; Köninger et al. 2022). Climate change and continued development on fire-prone landscapes will increase impacts of wildfires, like high suppression costs and loss of lives and properties (Bowman et al., 2017). the climate change effects on wildfire frequency and devastating effects considering the future climate conditions with the prolonged dry and warm summer period that favor the ignition and spread of wildfires (Goss et al. 2020),However, even in the highly fire-prone ecosystems, loss of biodiversity, ecosystem function or services after wildfire events occurring with unnaturally high frequency, magnitude of extent or intensity can result to land degradation or even complete transformation of the ecosystem. Besides their impacts on the carbon cycle, such events, usually called as megafires because of their size, reduce the amount of living biomass, affect species composition, water and nutrient cycles, increase flood risk and soil erosion, and threaten local livelihoods by burning agricultural land and homes. In addition, these fires have devastating impacts on local wildlife as animals either is unable to escape from the fires or become threatened by loss of their habitat, food and shelter.

Not only climatic conditions but also human activities influence fire regimes through their effects on ignition sources and fuel characteristics in many parts of the world. The tendency of the urbanization nearby or within forest ecosystems is a worldwide phenomenon that increases every year in Europe (Moreno et al., 2014), but also in the United States (Radeloff et al., 2018), Canada (Johnston and Flannigan, 2018; Parisien et al., 2020), and in Chile (Sarricolea et al., 2020),  Argentina in South America (Argañaraz et al., 2017). This incrising areas are what is called the wildland–urban interface (WUI). The WUI is the area where human-built structures and infrastructure abut or mix with naturally occurring vegetation types. Wildfires are of particular concern in the WUI because these areas comprise extensive flammable vegetation, numerous structures, and ample ignition sources.(Rutherford et al., 2010). Fires at the WUI are becoming increasingly hazardous for life safety and property protection (Vacca, et al., 2020). The combination of the aforementioned condition convert wildfire to megafires. Megafire is called an extraordinary fire that devastates a large area. They are notable for their physical characteristics including intensity, size, duration, and uncontrollable dimension, as well as their social characteristics, including suppression cost, damages and fatalities (Rodriguez et al., 2020).

Rodriguez, B., Lareau, N. P.,Kingsmill, D. E., &Clements, C. B.(2020).Extreme pyroconvective updrafts during a megafire.Geophysical Research Letters,47, e2020GL089001.https://doi.org/10.1029/2020GL089001

Stefanidis, S., Alexandridis, V., Spalevic, V., & Mincato, R. L. (2022). Wildfire Effects on Soil Erosion Dynamics: The Case Of 2021 Megafires in Greece. Agriculture & Forestry, 68(2), 49-63.

Efthimiou, N., Psomiadis, E., & Panagos, P. (2020). Fire severity and soil erosion susceptibility mapping using multi-temporal Earth Observation data: The case of Mati fatal wildfire in Eastern Attica, Greece. Catena, 187, 104320.

Köninger, J., Panagos, P., Jones, A., Briones, M. J. I., & Orgiazzi, A. (2022). In defence of soil biodiversity: Towards an inclusive protection in the European Union. Biological Conservation, 268, 109475.

Stefanidis, S., Alexandridis, V., & Ghosal, K. (2022). Assessment of Water-Induced Soil Erosion as a Threat to Natura 2000 Protected Areas in Crete Island, Greece. Sustainability, 14(5), 2738.

Bowman, D. M. J. S., Williamson, G. J., Abatzoglou, J. T., Kolden, C. A., Cochrane, M. A., and Smith, A. M. S. (2017). Human exposure and sensitivity to globally extreme wildfire events.Nat. Ecol. Evol.1:0058. doi: 10.1038/s41559-016-0058

Moreno, M. V., Conedera, M., Chuvieco, E., and Pezzatti, G. B. (2014). Fire regime changes and major driving forces in Spain from 1968 to 2010. Environ. Sci. Policy 37, 11–22. doi: 10.1016/j.envsci.2013.08.005

Radeloff, Volker C.; Helmers, David P.; Kramer, H. Anu; Mockrin, Miranda H.; Alexandre, Patricia M.; Bar-Massada, Avi; Butsic, Van; Hawbaker, Todd J.; Martinuzzi, Sebastián; Syphard, Alexandra D.; Stewart, Susan I. 2018. Rapid growth of the US wildland-urban interface raises wildfire risk. Proceedings of the National Academy of Sciences.https://doi.org/10.1073/pnas.1718850115.

Johnston, L. M., and Flannigan, M. D. (2018). Mapping Canadian wildland fire interface areas.Int. J. Wildl. Fire27, 1–14. doi: 10.1071/WF16221

Parisien, M., Barber, Q. E., Hirsch, K. G., Wang, X., Arseneault, D., Parks, S. A., et al. (2020). Fire Defity Increases Fire Risk for Many Communities in the Canadian Boreal Forest.Nat. Commun.11:2121. doi: 10.1038/s41467-020-15961-y

Sarricolea, P., Serrano-Notivoli, R., Fuentealba, M., Hernández-Mora, M., de la Barrera, F., Smith, P., et al. (2020). Recent wildfires in Central Chile: Detecting links between burned areas and population exposure in the wildland urban interface. Sci. Total Environ. 706:135894. doi: 10.1016/j.scitotenv.2019.135894

Argañaraz, J., Radeloff, V. C., Bar-Massada, A., Gavier-Pizarro, G. I., Scavuzzo, C. M., and Bellis, L. M. (2017). Assessing wild fire exposure in the Wildland-Urban Interface area of the mountains of central Argentina. J. Environ. Manage.196, 499–510. doi: 10.1016/j.jenvman.2017.03.058

Rutherford V. Platt, The Wildland–Urban Interface: Evaluating the Definition Effect, Journal of Forestry, Volume 108, Issue 1, January 2010, Pages 9–15, https://doi.org/10.1093/jof/108.1.9

Vacca, P., Caballero, D., Pastor, E., Planas, E. (2020) WUI fire risk mitigation in Europe: A performance-based design approach at home-owner level, Journal of Safety Science and Resilience,Volume 1, Issue 2, Pages 97-105, ISSN 2666-4496,https://doi.org/10.1016/j.jnlssr.2020.08.001.

  • Materials and Methods

One remark to Figure 1.

Please clarify, how operators can serve protected cabinet in everyday work activity?

Answer: The following text was added in lines 130-132 of the revised manuscript. “The panels were installed fixed to the cabinet with metallic screws and washers creating a space air gap of 5 cm, allowing the normal work activity, specially the door opening.”

  • Figures 2 should be large. This is more suitable for readers.

Answer: We agree with the comment. Figures 2 and 3 were enlarged to fit the page width.

  • Table 2. Please clarify, why did you choose wind speed in the range from 0 to 3 meter per second? Could you provide any supporting references or considerations?

Answer: The following text was added to the article to clarify the choose of the wind speed (lines 144-151of the revised manuscript:

The wind speed varied in the range from 0 to 3 m.s-1. These are standard values used in fire experiments in wind tunnels according to Abouali  et al. 2021 and Ribeiro et al. 2022. The flow over the floor of the tunnel is of a boundary layer type with a reference velocity Uo that is measured at the centre of the working section floor and 0.5 m above the ground this corresponds to a freestream at 10m height standard wind readings of 1.5xUo. The fire experiments were performed with the following values of Uo: 0,1, 2 and 3ms-1, that corresponds to winds at 10m height of 0, 1.5, 3.0, 4.5 ms-1 (0, 5.4, 10.8 to 16.2 km.h-1).

Abouali, A., Viegas, D. X., & Raposo, J. R. (2021). Analysis of the wind flow and fire spread dynamics over a sloped–ridgeline hill. Combustion and Flame, 234. https://doi.org/10.1016/j.combustflame.2021.111724

Ribeiro, C., Reis, L., Raposo, J., Rodrigues, A., Viegas, D. X., & Sharples, J. (2022). Interaction between two parallel fire fronts under different wind conditions. International Journal of Wildland Fire, 31(5), 492–506. https://doi.org/10.1071/WF21120

  • Please, provide characteristics of cameras Sony FDR-AX53 and Sony HXR-NX30E. This is more suitable for readers.

Answer: The main characteristics of cameras Sony FDR-AX53, Sony HXR-NX30E and InfraRed FLIR Camera SC660 were added to the manuscript (lines 164-175 of the revised manuscript).

  • Could you provide supporting references or consideration for your quantuity and positions of thermocouples. It seems to me you need more thermocouples to take into account spatial inhomogenity of forest fire front.

Answer: The increase in the temperature of the protection/cabinet is due to the heat received on the surfaces by thermal convection and mainly by thermal radiation from the entire fire front (which is much wider than the cabinet). For this reason, since the radiation received results from the integration over the entire fire front, it does not seem necessary to measure the temperature of the protection/cabinet at several points. Furthermore, the thermal inertia of the protection/cabinet materials contributes to temperature homogenization. Regarding the position of the thermocouples, they were placed, whenever possible, in the points or planes of symmetry, in order to reduce the effect of the finite width of the fire front.

  • Results

I did not find confidential intervals in the figures with results. Sorry, but you must provide information about uncertainty and error analysis. You should analyze both apparatus and random uncertainty. At present time your results looks like poor soundness figures. Please, provide description of uncertainty analysis procedure in materials and methods section and confidential intervals in all Figures with results.

Answer: Thank you very much for the reviewer's request for accuracy in the confidential intervals in the figures with results. We improved the figures adding the uncertainty intervals of the measurements and the following text was added to the article: (lines 203-229):

In equation (1), Φ [W/m2] is the heat flux, U [V] is the output voltage that is read directly from the signal generated by the heat flux sensor, S [V/(W/m2)] is the sensitivity of the sensor at 20ºC and  and T [ºC] is the temperature read on thermocouple of the sensor. The sensitivity of the sensor is available at the sensor calibration certificate and takes the value of 9.83x10-9 V/(W/m2) with a calibration uncertainty of ± 0.98x10-9 V/(W/m2). As referred to in the calibration certificate, this calibration uncertainty corresponds to the expanded uncertainty with a coverage factor k=2, and defines an interval estimated to have a level of confidence of 95%.

The expanded uncertainty of the heat flux measurements, for a confidence level of 95% was calculated according to [26], taking into account the main sources of uncertainty, namely: calibration uncertainty of the heat flux sensor; uncertainty due to the input noise error of NI 9211 board and  uncertainty due to the NI 9211 board sensitivity. The uncertainty due to the systematic error of the NI 9211 board was neglected since the systematic error was adequately corrected. The uncertainty due to the accuracy of the thermocouple of the sensor was neglected as the order of magnitude is lower in comparison to the others uncertainties. Consulting the NI 9211 board specifications [27], a 1 mV input noise  and 1 mV board sensitivity were considered.

The expanded uncertainty of the temperature measurements, for a confidence level of 95% was calculated according to [26], taking into account the main sources of uncertainty, namely: uncertainty due to the thermocouple accuracy, uncertainty due to the NI 9213 board error when connected to k type thermocouples and uncertainty due to the NI 9213 board sensitivity. Consulting the thermocouple specifications [28], the K type thermocouple accuracy was calculated by the function max(1.5ºC;0.004*temp[ºC]). Consulting the NI 9213 board specifications [29], the maximum error of the NI 9213 board at room temperature when connected to  k type thermocouples, for our  temperature range (20ºC<T<800 ºC), is 1.5 ºC and the board sensitivity is 0.25 ºC.

Reviewer 2 Report

The introduction must be enlarge and better state the problem

In the start of the introduction

Highlight the importance devastating effects of wildfires in the Mediterranean region.

Fire is considered as environmental factor acting in the Mediterranean climate, having played an obvious evolutionary role in the structure and function of Mediterranean climate ecosystems. In the aftermaths of the wildfire accelerated erosion occurs (Stefanidis et al. 2022a, Efthimiou et al. 2018) threaten the natural regeneration process and biodiversity and biotic natural capital recovery (Stefanidis et al. 2022b; Köninger et al. 2022). However, even in the highly fire-prone ecosystems, loss of biodiversity, ecosystem function or services after wildfire events occurring with unnaturally high frequency, magnitude of extent or intensity can result to land degradation or even complete transformation of the ecosystem. Besides their impacts on the carbon cycle, such events, usually called as megafires because of their size, reduce the amount of living biomass, affect species composition, water and nutrient cycles, increase flood risk and soil erosion, and threaten local livelihoods by burning agricultural land and homes. In addition, these fires have devastating impacts on local wildlife as animals either is unable to escape from the fires or become threatened by loss of their habitat, food and shelter.

Stefanidis, S., Alexandridis, V., Spalevic, V., & Mincato, R. L. (2022a). Wildfire Effects on Soil Erosion Dynamics: The Case Of 2021 Megafires in Greece. Agriculture & Forestry, 68(2), 49-63.

Efthimiou, N., Psomiadis, E., & Panagos, P. (2020). Fire severity and soil erosion susceptibility mapping using multi-temporal Earth Observation data: The case of Mati fatal wildfire in Eastern Attica, Greece. Catena, 187, 104320.

Köninger, J., Panagos, P., Jones, A., Briones, M. J. I., & Orgiazzi, A. (2022). In defence of soil biodiversity: Towards an inclusive protection in the European Union. Biological Conservation, 268, 109475.

Stefanidis, S., Alexandridis, V., & Ghosal, K. (2022). Assessment of Water-Induced Soil Erosion as a Threat to Natura 2000 Protected Areas in Crete Island, Greece. Sustainability, 14(5), 2738.

Also, some statements about the about the climate change effects on wildfire frequency and devastating effects considering the future climate conditions with the prolonged dry and warm summer period that favor the ignition and spread of wildfires (Goss et al. 2020), as well as the increase in wildland–urban interface (WUI) areas, as housing expands in and near forests due to the pressure for urban areas (Radeloff et al. 2018). The combination of the aforementioned condition convert wildfire to megafires. Megafire is called an extraordinary fire that devastates a large area. They are notable for their physical characteristics including intensity, size, duration, and uncontrollable dimension, as well as their social characteristics, including suppression cost, damages and fatalities (Buckland 2019).

In the last paragraph of the Indroduction clearly state the novelty points of the current approach and the research gap answered from this research.

The discussion is rather small. The results should be discussed in a wider context and comparison with previous or similar researches should be done. Also, some targets for future research must be added and explained why.  

Author Response

Responses to Reviewer 2

Manuscript ID: 1832472

Title: Rigid protection system of infrastructures against forest fires

Authors: Gilberto Vaz * , Jorge Rafael Raposo , Luís Duarte Reis , Pedro Monteiro, Domingos Xavier Viegas

The authors would like to thank the reviewer for the constructive comments. The comments and concerns raised have been duly addressed in this revised version of the paper.

Responses to reviewer review are detailed described below. In the revised manuscript, the revised sentences are inserted with track changes.

Reviewer#2

  • The introduction must be enlarge and better state the problem

In the start of the introduction

Highlight the importance devastating effects of wildfires in the Mediterranean region. 

Fire is considered as environmental factor acting in the Mediterranean climate, having played an obvious evolutionary role in the structure and function of Mediterranean climate ecosystems.In the aftermaths of the wildfire accelerated erosion occurs (Stefanidis et al. 2022a, Efthimiou et al. 2018) threaten the natural regeneration process and biodiversity and biotic natural capital recovery (Stefanidis et al. 2022b; Köninger et al. 2022).However, even in the highly fire-prone ecosystems, loss of biodiversity, ecosystem function or services after wildfire events occurring with unnaturally high frequency, magnitude of extent or intensity can result to land degradation or even complete transformation of the ecosystem. Besides their impacts on the carbon cycle, such events, usually called as megafires because of their size, reduce the amount of living biomass, affect species composition, water and nutrient cycles, increase flood risk and soil erosion, and threaten local livelihoods by burning agricultural land and homes. In addition, these fires have devastating impacts on local wildlife as animals either is unable to escape from the fires or become threatened by loss of their habitat, food and shelter.

Stefanidis, S., Alexandridis, V., Spalevic, V., & Mincato, R. L. (2022a). Wildfire Effects on Soil Erosion Dynamics: The Case Of 2021 Megafires in Greece. Agriculture & Forestry, 68(2), 49-63.

Efthimiou, N., Psomiadis, E., & Panagos, P. (2020). Fire severity and soil erosion susceptibility mapping using multi-temporal Earth Observation data: The case of Mati fatal wildfire in Eastern Attica, Greece. Catena, 187, 104320.

Köninger, J., Panagos, P., Jones, A., Briones, M. J. I., & Orgiazzi, A. (2022). In defence of soil biodiversity: Towards an inclusive protection in the European Union. Biological Conservation, 268, 109475.

Stefanidis, S., Alexandridis, V., & Ghosal, K. (2022). Assessment of Water-Induced Soil Erosion as a Threat to Natura 2000 Protected Areas in Crete Island, Greece. Sustainability, 14(5), 2738.

Also, some statements about the about the climate change effects on wildfire frequency and devastating effects considering the future climate conditions with the prolonged dry and warm summer period that favor the ignition and spread of wildfires (Goss et al. 2020), as well as the increase in wildland–urban interface (WUI) areas, as housing expands in and near forests due to the pressure for urban areas (Radeloff et al. 2018). The combination of the aforementioned condition convert wildfire to megafires. Megafire is called an extraordinary fire that devastates a large area. They are notable for their physical characteristics including intensity, size, duration, and uncontrollable dimension, as well as their social characteristics, including suppression cost, damages and fatalities (Buckland 2019). 

Answer: Thank you. We agree with your comment and we revised the text. The following text and references were added according the requirements of reviewer 1 and 2 (lines 30-107 of revised document):

Fire is considered as environmental factor acting in the Mediterranean climate, having played an obvious evolutionary role in the structure and function of Mediterranean climate ecosystems. In the aftermaths of the wildfire accelerated erosion occurs (Stefanidis et al. 2022a, Efthimiou et al. 2018) threaten the natural regeneration process and biodiversity and biotic natural capital recovery (Stefanidis et al. 2022b; Köninger et al. 2022). Climate change and continued development on fire-prone landscapes will increase impacts of wildfires, like high suppression costs and loss of lives and properties (Bowman et al., 2017). the climate change effects on wildfire frequency and devastating effects considering the future climate conditions with the prolonged dry and warm summer period that favor the ignition and spread of wildfires (Goss et al. 2020),However, even in the highly fire-prone ecosystems, loss of biodiversity, ecosystem function or services after wildfire events occurring with unnaturally high frequency, magnitude of extent or intensity can result to land degradation or even complete transformation of the ecosystem. Besides their impacts on the carbon cycle, such events, usually called as megafires because of their size, reduce the amount of living biomass, affect species composition, water and nutrient cycles, increase flood risk and soil erosion, and threaten local livelihoods by burning agricultural land and homes. In addition, these fires have devastating impacts on local wildlife as animals either is unable to escape from the fires or become threatened by loss of their habitat, food and shelter.

Not only climatic conditions but also human activities influence fire regimes through their effects on ignition sources and fuel characteristics in many parts of the world. The tendency of the urbanization nearby or within forest ecosystems is a worldwide phenomenon that increases every year in Europe (Moreno et al., 2014), but also in the United States (Radeloff et al., 2018), Canada (Johnston and Flannigan, 2018; Parisien et al., 2020), and in Chile (Sarricolea et al., 2020),  Argentina in South America (Argañaraz et al., 2017). This incrising areas are what is called the wildland–urban interface (WUI). The WUI is the area where human-built structures and infrastructure abut or mix with naturally occurring vegetation types. Wildfires are of particular concern in the WUI because these areas comprise extensive flammable vegetation, numerous structures, and ample ignition sources.(Rutherford et al., 2010). Fires at the WUI are becoming increasingly hazardous for life safety and property protection (Vacca, et al., 2020). The combination of the aforementioned condition convert wildfire to megafires. Megafire is called an extraordinary fire that devastates a large area. They are notable for their physical characteristics including intensity, size, duration, and uncontrollable dimension, as well as their social characteristics, including suppression cost, damages and fatalities (Rodriguez et al., 2020).

Rodriguez, B., Lareau, N. P.,Kingsmill, D. E., &Clements, C. B.(2020).Extreme pyroconvective updrafts during a megafire.Geophysical Research Letters,47, e2020GL089001.https://doi.org/10.1029/2020GL089001

Stefanidis, S., Alexandridis, V., Spalevic, V., & Mincato, R. L. (2022). Wildfire Effects on Soil Erosion Dynamics: The Case Of 2021 Megafires in Greece. Agriculture & Forestry, 68(2), 49-63.

Efthimiou, N., Psomiadis, E., & Panagos, P. (2020). Fire severity and soil erosion susceptibility mapping using multi-temporal Earth Observation data: The case of Mati fatal wildfire in Eastern Attica, Greece. Catena, 187, 104320.

Köninger, J., Panagos, P., Jones, A., Briones, M. J. I., & Orgiazzi, A. (2022). In defence of soil biodiversity: Towards an inclusive protection in the European Union. Biological Conservation, 268, 109475.

Stefanidis, S., Alexandridis, V., & Ghosal, K. (2022). Assessment of Water-Induced Soil Erosion as a Threat to Natura 2000 Protected Areas in Crete Island, Greece. Sustainability, 14(5), 2738.

Bowman, D. M. J. S., Williamson, G. J., Abatzoglou, J. T., Kolden, C. A., Cochrane, M. A., and Smith, A. M. S. (2017). Human exposure and sensitivity to globally extreme wildfire events.Nat. Ecol. Evol.1:0058. doi: 10.1038/s41559-016-0058

Moreno, M. V., Conedera, M., Chuvieco, E., and Pezzatti, G. B. (2014). Fire regime changes and major driving forces in Spain from 1968 to 2010. Environ. Sci. Policy 37, 11–22. doi: 10.1016/j.envsci.2013.08.005

Radeloff, Volker C.; Helmers, David P.; Kramer, H. Anu; Mockrin, Miranda H.; Alexandre, Patricia M.; Bar-Massada, Avi; Butsic, Van; Hawbaker, Todd J.; Martinuzzi, Sebastián; Syphard, Alexandra D.; Stewart, Susan I. 2018. Rapid growth of the US wildland-urban interface raises wildfire risk. Proceedings of the National Academy of Sciences.https://doi.org/10.1073/pnas.1718850115.

Johnston, L. M., and Flannigan, M. D. (2018). Mapping Canadian wildland fire interface areas.Int. J. Wildl. Fire27, 1–14. doi: 10.1071/WF16221

Parisien, M., Barber, Q. E., Hirsch, K. G., Wang, X., Arseneault, D., Parks, S. A., et al. (2020). Fire Defity Increases Fire Risk for Many Communities in the Canadian Boreal Forest.Nat. Commun.11:2121. doi: 10.1038/s41467-020-15961-y

Sarricolea, P., Serrano-Notivoli, R., Fuentealba, M., Hernández-Mora, M., de la Barrera, F., Smith, P., et al. (2020). Recent wildfires in Central Chile: Detecting links between burned areas and population exposure in the wildland urban interface. Sci. Total Environ. 706:135894. doi: 10.1016/j.scitotenv.2019.135894

Argañaraz, J., Radeloff, V. C., Bar-Massada, A., Gavier-Pizarro, G. I., Scavuzzo, C. M., and Bellis, L. M. (2017). Assessing wild fire exposure in the Wildland-Urban Interface area of the mountains of central Argentina. J. Environ. Manage.196, 499–510. doi: 10.1016/j.jenvman.2017.03.058

Rutherford V. Platt, The Wildland–Urban Interface: Evaluating the Definition Effect, Journal of Forestry, Volume 108, Issue 1, January 2010, Pages 9–15, https://doi.org/10.1093/jof/108.1.9

Vacca, P., Caballero, D., Pastor, E., Planas, E. (2020) WUI fire risk mitigation in Europe: A performance-based design approach at home-owner level, Journal of Safety Science and Resilience,Volume 1, Issue 2, Pages 97-105, ISSN 2666-4496,https://doi.org/10.1016/j.jnlssr.2020.08.001.

  • In the last paragraph of the Indroduction clearly state the novelty points of the current approach and the research gap answered from this research.

Answer: Thank you. We agree with your comment. The following text was added to the end of the introduction (lines 101-107 of the revised manuscript):

The novelty of the protection is shown by its effectiveness to the protection of telecommunication cabinets and other similar infrastructures against forest fires even in extreme conditions of fire with wind. The protection avoided the high temperatures in the cabinet. Without protection a very expensive and critical system can be easily damaged by the fire front of a wildfire. Another great advantage of this protection is its low cost of material and the reduced labour for installation allowing the daily work on the cabinet if necessary.

  • The discussion is rather small. The results should be discussed in a wider context and comparison with previous or similar researches should be done.

Answer: Thank you. The following text was added to the discussion (lines 287-319 of the revised manuscript):

Figures 6 and 7 show that for the higher laboratory wind speed, the heat absorbed by the protection is significant, even in these laboratory tests, since the outer front surface temperatures of the protection reach 180-190ºC (thermocouple T_3). Figure 8 shows that, without protection, the outer front surface of the cabinet also reaches approx. 170ºC (thermocouple T_3). This lower temperature in comparison to the other tests with protection installed (Test 03 and 06) can be explained as a consequence of the higher thermal conductivity of the metallic cabinet material.

Figures 6 and 7 also show clearly the thermal behaviour of the protection for the case of higher laboratory wind speed. It can be observed that the inner front surface temperatures of the protection (thermocouple T_2) reach 60-70ºC. Comparing these temperatures to the outer front surface temperatures of the protection (thermocouple T_3),  180-190ºC as referred above, it is clear the effect of the low thermal conductivity and thermal inertia of the protection, as expected.

Figure 8 shows that the heat transfer problem studied is clearly a transient phenomenon since the temperature inside the cabinet (thermocouple T_1) increases, even when the other temperatures are already decreasing. This is why it is important to install a protection with a high thermal inertia.

Figures 9, 10 and Table 3 shows that the sidewalls are important. Comparing Figures 9 and 10, it is observed that the inexistence of side walls (test 03), allowing lateral air entrainment with significant edge effect, generates a more curved fire front, with a higher rate of spread along the symmetry line of the fuel bed. On the contrary, when the sidewalls are installed (test 06), the lateral air entrainment is reduced, the fire front is more linear and the rate of spread along the symmetry line of the fuel bed is slightly lower. The values for the maximum dimensionless rate of spread are presented in Table 3 (test 03 - 18.1; test 06 - 14.96).

Table 3 shows that the protection is effective, as the maximum temperature inside the cabinet remains below 30 °C.  Even under high fireline intensities (tests 03 and 06) the temperature evolution in two tests performed with the protection applied to the telecommunications cabinet and 3 m.s-1 wind speed (highest wind tunnel speed which is the most critical situation tested and corresponding to the typical ground level wind speeds of intense forest fires). These results are better compared to the ones obtained in similar tests using a non-rigid protection of fibreglass thin blankets with an aluminium coating tested in the same cabinets by [21].

Additionally, Figures 9 and 10 were added to enrich the results section, showing the fire front shape evolution for a 5s time step for tests 03 and 06. (lines 266-274 of the revised manuscript)

  • Also, some targets for future research must be added and explained why.  

Answer: Thank you. The following future research was pointed (lines 347-351 of the revised manuscript):

As future work a physical and mathematical model shall be developed to accurately reproduce the heat transfer phenomena and temperature variation in the protection and cabinet, according to the incident heat flux, that also allows to study the effects of material, thickness and shape of protection.

Also, it will be important to test this type of protection under real fires, at field scale.

Round 2

Reviewer 1 Report

Dear authors!

Thank you for the revised version of your manuscript.

You addressed the majority of my comments.

But I suggest to increase number of cited works reaching about 50 references in total.

Also I ask you to clarify situation with random uncertainty.

You refered to K-thermocouple description on error measurement of temperature.

But this is apparatus uncertainty.

How many times you proceed experimental procedure with rigid protection.

You must understand that single measurement can not be used to make any conclusions on random uncertainty.

Please clarify this situation. So you may obtain large confidential intervals according multiple measurements of temperature in rigid protection within the same conditions. Intervals provided by you is very small. Maybe you use apparatus uncertainty to mark out such intervals.

I suggest major revision.

Author Response

Responses to Reviewer 1

Manuscript ID: 1832472

Title: Rigid protection system of infrastructures against forest fires

Authors: Gilberto Vaz * , Jorge Rafael Raposo , Luís Duarte Reis , Pedro Monteiro, Domingos Xavier Viegas

The authors would like to thank the reviewer for the constructive comments. The comments and concerns raised have been duly addressed in this revised version of the paper.

Responses to reviewer review are detailed described below. In the revised manuscript, the revised sentences are inserted with track changes.

Reviewer#1

Dear authors!

Thank you for the revised version of your manuscript.

You addressed the majority of my comments.

  • But I suggest to increase number of cited works reaching about 50 references in total.

Answer: Thank you very much for your suggestion. We improved the document, increasing significantly the number of cited works (above 50 references in total).

  • Also I ask you to clarify situation with random uncertainty.

You refered to K-thermocouple description on error measurement of temperature.

But this is apparatus uncertainty.

How many times you proceed experimental procedure with rigid protection.

You must understand that single measurement can not be used to make any conclusions on random uncertainty.

Please clarify this situation. So you may obtain large confidential intervals according multiple measurements of temperature in rigid protection within the same conditions. Intervals provided by you is very small. Maybe you use apparatus uncertainty to mark out such intervals.

I suggest major revision.

Answers: Thank you very much for the reviewer's request to clarify the calculation of the expanded uncertainty intervals. We agree with your comments, and we clarify that the expanded uncertainty calculated and showed in the figures is the apparatus uncertainty. The word “apparatus” was added to the text: (lines 257, 267, 281 and 305).

Random uncertainty resulting from multiple tests within the same experimental conditions was not considered because a single test was performed for each type of test. To clarify the number of experimental tests performed for each test type, the following text was added (lines 202-207): “These types of tests are very time consuming and require a lot of laboratory equipment. Preliminary tests were carried out to assess the need to repeat the tests for the purpose of evaluating the effectiveness of the thermal protection. Since the recordings of temperatures and heat fluxes were consistent, showed a stable evolution without significant random oscillations, a single test was performed for each type of test.”

Reviewer 2 Report

The article revized according to the reviewer comments and now can be accepted 

Author Response

Responses to Reviewer 2

Manuscript ID: 1832472

Title: Rigid protection system of infrastructures against forest fires

Authors: Gilberto Vaz * , Jorge Rafael Raposo , Luís Duarte Reis , Pedro Monteiro, Domingos Xavier Viegas

Responses to reviewer review are detailed described below.

Reviewer#2

  • The article revized according to the reviewer comments and now can be accepted

Answer: Thank you very much.
